# Transcriptome and Degradome Profiling Reveals a Role of miR530 in the Circadian Regulation of Gene Expression in *Kalanchoë marnieriana*

**DOI:** 10.3390/cells10061526

**Published:** 2021-06-17

**Authors:** Zhikang Hu, Ziyan Nie, Chao Yan, Hu Huang, Xianjin Ma, Yupeng Wang, Ning Ye, Gerald A. Tuskan, Xiaohan Yang, Hengfu Yin

**Affiliations:** 1State Key Laboratory of Tree Genetics and Breeding, Research Institute of Subtropical Forestry, Chinese Academy of Forestry, Hangzhou 311400, China; huzhikang01@163.com (Z.H.); nzy1211@gmail.com (Z.N.); 15272978041@163.com (H.H.); 15138558389@163.com (X.M.); 2College of Information Science and Technology, Nanjing Forestry University, Nanjing 210037, China; yupengwang@njfu.edu.cn (Y.W.); yening@njfu.edu.cn (N.Y.); 3Key Laboratory of Forest Genetics and Breeding, Research Institute of Subtropical Forestry, Chinese Academy of Forestry, Hangzhou 311400, China; 4Experimental Center for Subtropical Forestry, Chinese Academy of Forestry, Fenyi 336600, China; yanc01@163.com; 5Biosciences Division, Oak Ridge National Laboratory, Oak Ridge, TN 37830, USA; tuskanga@ornl.gov (G.A.T.); yangx@ornl.gov (X.Y.); 6DOE-Center for Bioenergy Innovation (CBI), Oak Ridge National Laboratory, Oak Ridge, TN 37830, USA

**Keywords:** crassulacean acid metabolism, microRNA, photosynthesis, drought, circadian clock, *Kalanchoë*

## Abstract

Crassulacean acid metabolism (CAM) is an important photosynthetic pathway for plant adaptation to dry environments. CAM plants feature a coordinated interaction between mesophyll and epidermis functions that involves refined regulations of gene expression. Plant microRNAs (miRNAs) are crucial post-transcription regulators of gene expression, however, their roles underlying the CAM pathway remain poorly investigated. Here, we present a study characterizing the expression of miRNAs in an obligate CAM species *Kalanchoë marnieriana*. Through sequencing of transcriptome and degradome in mesophyll and epidermal tissues under the drought treatments, we identified differentially expressed miRNAs that were potentially involved in the regulation of CAM. In total, we obtained 84 miRNA genes, and eight of them were determined to be *Kalanchoë*-specific miRNAs. It is widely accepted that CAM pathway is regulated by circadian clock. We showed that miR530 was substantially downregulated in epidermal peels under drought conditions; miR530 targeted two tandem zinc knuckle/PLU3 domain encoding genes (TZPs) that were potentially involved in light signaling and circadian clock pathways. Our work suggests that the miR530-TZPs module might play a role of regulating CAM-related gene expression in *Kalanchoë*.

## 1. Introduction

Crassulacean acid metabolism (CAM) is a modification of the photosynthesis pathway involving the cooperation of multiple processes, including carbon metabolism and circadian reversion of stomatal movements [1,2]. In CAM plants, the Calvin cycle is prefixed by initial incorporation of CO_2_ into malate during the night, then during daytime, the stomata are closed, and concentrated CO_2_ is released to ribulose bisphosphate carboxylase oxygenase (Rubisco) for fixation [1,2]. This diurnal rearrangement attenuates the losses of water and enhances the efficiency of photosynthesis by reducing photorespiration [3]; hence, many CAM plants are classified as having high water use efficiency (WUE) and survive under extremely dry environments [4,5].

The adaptation of CAM to dry environments is thought to be driven by carbon concentration mechanisms and has evolved numerous times in independent plant lineages [6,7,8]. CAM resembles the C4 pathway in the process of the separation of CO_2_ fixation and Rubisco reaction [9], while CAM plants do not require anatomical changes to chloroplasts [5,7]. The continuum system of CAM evolution (i.e., cycling, weak, idling, strong) suggests that modification of regulatory pathways to accomplish the C3-to-CAM conversion is achievable [9,10]. Recently, engineering the CAM pathway into C3 crops was proposed to be a potent resolution to improve the sustainability of global agriculture and agroforestry [11,12,13,14]. That is, by increasing water use efficiency (WUE), the engineered herbaceous and woody crops are expected to be more resilient and competent in warmer and drier environments [10,11,12,15].

To uncover the molecular drivers of CAM, extensive research, using large-scale transcriptome, metabolome and proteome, has been reported [16]. A wealth of genome information, gene expression databases and functional genomics tools are now available in CAM plants [17] and new molecular players have been identified as important regulators in the control of CAM processes, e.g., circadian clock genes and organic acid transporters [18,19,20]. With the support of high-quality reference genomes, functional genomics research in CAM model systems of different plant lineages, has been conducted [17]. *Kalanchoë* is among the best models due to the abundant knowledge of CAM physiology and biochemistry, the facile transformation and genome editing system, and the high-quality reference genome [17,19,21,22]. Particularly, the knockdown lines of NAD-malic enzyme (NAD-ME), cytosolic/plastidic pyruvate orthophosphate dikinase (PPDK) and phosphoenolpyruvate carboxylase kinase (PPCK) of *K. fedtschenkoi* have displayed impaired rhythms of CO_2_ fixation and the temporal expression patterns of molecular clock genes, providing evidence for circadian clock-mediated regulation of CAM processes [23,24]. Recently, the CRISPR/Cas9-mediated mutagenesis has also been carried out in *K. fedtschenkoi*, which provides a valuable platform for functional studies of CAM [22].

Plant microRNAs (miRNAs) are essential post-transcriptional regulators that are involved in various biological processes including development, growth and stress responses [25,26]. Additionally, many miRNAs are responsive and functionally specialized to biotic and abiotic stresses [27,28]. Evidence indicates that some deeply originating miRNAs families are conserved in regulating multiple stress-related processes [27]. Additionally, the miRNAs displaying diurnal expression patterns in Arabidopsis have been reported, suggesting that they play regulatory roles in circadian clock and light signaling [29]. Whether miRNAs play a role in the regulation of CAM-related biological processes (e.g., leaf succulence, drought resistance and circadian clock) remains unknown. With the advent of high-throughput sequencing technologies, miRNAs have been characterized in several CAM species. A recent analysis of transcriptome and miRNAome in pineapple (an obligate CAM species) revealed miRNAs with diel expression patterns and pointed to some interesting miRNA-target regulations relevant to the CAM pathway [30]. Still, little information of drought-responsive miRNAs is known in any CAM plant. Here, we performed small RNA, transcriptome and degradome sequencing analyses in *K. marnieriana* under drought conditions. We compared the miRNA expression profiles between epidermis and mesophyll tissues and revealed several deeply conserved as well as *Kalanchoë*-specific miRNAs were responsive to drought stress. We uncovered that miR530 displays a circadian pattern of expression and targets two tandem zinc knuckle/PLU3 domain encoding genes (TZPs), suggesting a potential role of regulating CAM-related gene expression.

## 2. Materials and Methods

### 2.1. Plant Materials

Mature plants of *K. marnieriana* were grown in the greenhouse of the Research Institute of Subtropical Forestry (119°57′ N, 30°04′ E; Fuyang, Zhejiang, China). Tissue was collected from propagated plants of *K. marnieriana* grown in a growth chamber under photoperiod of 12 h light/12 h dark at 24 °C and 40% humidity. Additionally, *K. marnieriana* plants were subjected to the drought treatments: 10 days without watering (D10), 25 days without watering (D25) and well-watered as the control condition. For the Agrobacterium-infiltration assay, the *Nicotiana benthamiana* plants were grown at 22 °C under 16 h light/8 h dark condition; one-month-old plants were used for the infiltration as described by [31].

### 2.2. RNA Preparation

For RNA preparation, the plant tissues were collected and frozen in liquid nitrogen and stored at −80 °C before subsequent uses. Total RNA was extracted using an RNAprep Pure Plant Kit (DP441, Tiangen, Beijing, China) from the mesophyll, epidermal peel and leaves of *K. marnieriana*. The concentration and integrity of the total RNA were checked before library construction. A Nanodrop 2000 spectrophotometer (Thermo Fisher, CA, USA) was used to calculate the RNA concentration, and samples with more than 200 ng/uL and optical density (OD) 260/280 above 2.0 were used for small RNA sequencing. Near equal amounts of RNAs from different tissue types were mixed to construct libraries for transcriptome assembly and degradome analysis.

### 2.3. RNA Sequencing and Analysis

The small RNA sequencing libraries were prepared by a TruSeq Small RNA Library Preparation Kits (Illumina, San Diego, CA, USA) according to user’s manual from Illumina as described by [32]. Briefly, following the isolation of suitable cDNA fragments by gel electrophoresis analysis, the obtained libraries were enriched with small RNAs and loaded onto an Illumina HiSeqTM 2000 platform for sequencing by Hangzhou LC-Bio Co., Ltd. (Hangzhou, China). Total reads were trimmed of low-quality and adapter sequences using the Illumina Pipeline. The mixed RNA sample was sequenced and assembled into nonredundant unigenes using Trinity2.1.1.0 (Cambridge, MA, USA) [33]. Unigenes were tentatively identified based on the best hits against known sequences in the database. The transcriptome data of *K. marnieriana* were available in National Center for Biotechnology Information (TSA accession number GIXV00000000). All sequencing reads were deposited into the NCBI SRA database under Bioproject PRJNA684436. The identification of miRNA was initially identified according previous pipelines [34]. Genome-wide prediction of miRNAs was performed using the Shortstack3.4 (University Park, PA, USA) [35] and combined results were obtained by removing redundant miRNAs with precursor sequences. To quantify the abundance of miRNA, a transcripts per million value was defined as ‘counts of read mapped to miRNA * 1,000,000’/‘reads mapped to reference genome’, followed by differential expression analysis as described by [36]. For the degradome analysis, the extracted sequencing reads over 20 nucleotides were used to identify potentially cleaved targets by the CleaveLand 3.0 pipeline (University Park, PA, USA), and *p*-value of less than 0.05 was used to identify highly confident targets [37]. The *Kalanchoë* genome references were used to identify the annotation of target genes.

### 2.4. Dual-Fluorescence Assay

To validate the miRNA target genes, the predicted target site and miRNA precursor sequences were constructed into the Dual-Luciferase Sensors system (Addgene, 55206, Watertown, MA, USA) [31]. All constructs were verified by sequencing. The miR530 precursor was obtained by amplifying the *K. marnieriana* genomic DNA using the sequence specific primers (Appendix A). Then, the miRNA overexpression construct was coinfiltrated with the target sites vector in tobacco accordingly [31]. For each combination, including control and test groups, at least five biological replicates were used for efficiency analysis [38].

### 2.5. Real-Time Quantification of Gene Expression

The expression levels of miRNAs were examined by qRT-PCR in conjunction with a Mir-X™ miRNA FirstStrand Synthesis kit (Cat. 638313, TaKaRa, Dalian, China) according to the user’s manual. The target gene expression analysis was performed using a PrimeScript kit (Cat. RR037Q, TaKaRa, Dalian, China). The gene-specific primers (Appendix A) were designed by Primer Express 3.0.1 (Applied Biosystems, Foster City, CA, USA). The RT-qPCR was performed on an ABI PRISM 7300 Real-Time PCR System (Foster City, CA, USA). For qRT-PCR experiments, three biological replicates were used, and each was repeated three times as technical replicates. The data were analyzed with the 2–ΔΔCT method [39].

## 3. Results

### 3.1. K. marnieriana Is an Obligate CAM Plant and Its CAM Expression Is Enhanced by Drought Treatment

*K. marnieriana* is a closely related species to *K. fedtschenkoi* and *K. laxiflora* according to its morphology and biological characteristics [40] (Appendix A). Using the sequence of 18S rRNA, we determined that this accession was closely related to other two reported *K. marnieriana* plants (Appendix A). To assess the drought responses in *K. marnieriana*, we performed the acidity titration measurements between dawn and dusk under different watering conditions. We found that, under the regularly watering condition, the difference value (ΔH+) of titratable acid in mature leaves of *K. marnieriana* averaged 72 nmol H+ per kg, and the acidity levels were significantly increased under the 10 days (D10) and 25 days (D25) of drought treatments (Appendix A), indicating that the drought treatments enhanced the production of leaf titratable acids.

To reveal the miRNAs and their targets that were responsive to drought, we designed an experiment of drought treatment by withholding water. Leaf epidermis and mesophyll tissues were separated and collected for sampling at the D10 and D25 conditions (Figure 1). We performed a combination of transcripts and small RNAs sequencing analyses and focused on the miRNAs and their target genes. The *K. laxiflora* genome and transcriptome assembly were both used as the references for miRNA gene identification. The transcriptome assembly was obtained by using the mixed RNA samples contained 26,336 unigenes with the N50 of 1434 bp (TSA GIXV00000000). We consolidated the prediction results to remove redundant miRNAs. In total, we obtained 94 miRNAs, of which 29 were supported by the genome assembly of *K. fedtschenkoi* (Figure 1; Appendix A). To predict the targets of miRNA, we performed the Parallel Analysis of RNA Ends (PARE, also known as degradome sequencing) to reveal the potential targeted transcripts of miRNAs in *K. marnieriana* (Figure 1). The potential targets of miRNA were identified based on the frequency of RNA ends. We determined that the integrative analyses of differential expression of miRNAs and their targets were sufficient to inform the CAM-related regulation of gene expression in *Kalanchoë*.

### 3.2. Identification of Conserved miRNAs and Lineage-Specific miRNAs in K. marnieriana

To uncover the miRNAs, we filtered the low-quality sequences and obtained 18–25 bp small RNA (sRNA) sequencing reads for the bioinformatics identification pipeline (Appendix A). For the miRNA identification from transcriptome assembly, the secondary structure and the reads-mapping profile were evaluated to generate the high-confident miRNAs. We found that the 29 genome-supported miRNA displayed a more uniform precursor length and higher expression levels comparing to the transcriptome-based miRNAs (Figure 2A). The mature sequences of miRNAs were annotated using plant miRNAs databases. We discovered that eight miRNAs from the 29 genome-supported miRNAs were not homologous to other plant miRNAs and were designated as *Kalanchoë*-specific miRNAs (Figure 2B). To further examine these *Kalanchoë*-specific miRNAs, we analyzed the secondary structure and sRNA coverage of precursors; all eight miRNAs displayed canonical features of miRNA genes (Figure 2B).

### 3.3. Differential Expression of miRNAs in Responsive to Drought Stress in Mesophyll and Epidermis

To identify miRNA genes responsive to drought, we performed expression analysis to reveal differentially expressed miRNAs. The expression level of mature miRNAs at each condition was quantified (Appendix A) and used for subsequent statistical analysis. We found that 55 miRNAs were differentially expressed in epidermis, which was more than twice the number in mesophyll (Figure 3A). Only four miRNAs were found in both tissues (Figure 3B); 28 miRNAs were discovered under the D25 condition in epidermis (Figure 3B). These results suggested that the drought stress might cause substantial changes of small RNAome, which could subsequently regulate the gene expression. We then focused on the *Kalanchoë*-specific miRNAs expression patterns under the drought conditions. We found that six *Kalanchoë*-specific miRNAs were responsive to the drought treatments (Figure 3C). Particularly, Kal-miR-C1878, significantly upregulated in both mesophyll and epidermis, and Kal-miR-C689, which showed opposite patterns between mesophyll and epidermis under the drought conditions (Figure 3C).

### 3.4. Diurnal Expression of the Targets of miRNAs in K. marnieriana Leaves

The diurnal expression of gene expression is fundamental to CAM, and we ask whether the differentially expressed miRNAs under drought stress were involved in the circadian expression of genes. Through the Parallel Analysis of RNA Ends (also known as PARE sequencing or degradome), we predicted the targets of miRNAs by using the transcripts from the *Kalanchoë* genome database. In total, 381 miRNA-target regulations (*p* < 0.05) were identified (Appendix A). To further search for the miRNA-target regulation related to CAM pathway, we evaluated the diurnal transcriptome dataset from K. *fedtschenkoi* leaves [19]. We focused on genes that displayed the expression peaks at morning and dusk periods (Appendix A). Interestingly, we found that *PLASMA MEMBRANE PROTON ATPASE 2* (Kaladp0008s0304) was identified as a target of miR369, which was revealed as a key regulator of stomatal movement in *K. fedtschenkoi* [19]. In the dusk-peaked gene cluster, we showed that two candidate targets of Kal-miR_C1817, including a MYB transcription factor and POF like gene, displayed high expression levels in the daytime (Appendix A). In the morning-peaked gene cluster, we found that two TZP-like genes were the targets of miR530 (Appendix A). TZP genes, also known as BLUS3-like genes, were essential regulators of circadian clock and blue light signaling pathways [41,42]. We hypothesized that the miR530-TZP regulatory module might be involved in directing the diurnal gene expression patterns in *Kalanchoë*.

### 3.5. The Expression of miR530 and TZP Genes Is Affected by Circadian Clock and Light Signaling

To investigate the miR530-TZP regulatory module, we identified and validated the genomic locus of miR530 through cloning and sequencing. The precursor of miR530 contains a typical stem-loop structure with abundant accumulation of small RNAs at the mature and star regions (Figure 4A). Based on the deep sequencing analysis, we found that the mature miR530 levels in mesophyll tissues were not significantly changed under the drought treatments (Figure 4B), while more than 20-fold reduction was revealed in the epidermis under D10 and D25 conditions (Figure 4C). To validate the targets of miR530, the Dual-Luciferase Transient Expression System was employed to evaluate the efficacy of miRNA and its targets [38]. This analysis showed that both targeting sites of *TZP1* and *TZP2* were regulated by miR530 (Figure 4C), suggesting those transcripts were both cleaved by miR530 in *Kalanchoë*.

In order to reveal the roles of miR530 in the regulation circadian gene expression, we performed time-course analysis of miR530, *TZP1* and *TZP2* expression. Under the normal light–dark cycling condition, we showed that the miR530 expression peaked in the middle of the night and dropped to the lowest level before dawn (Figure 5A); this pattern negatively correlated with expression of *TZP1/2* which peaked before the onset of light (Figure 5A). The expression pattern of miR530 under normal light–dark cycle was tested by JTK_cycle, which displayed a rhythmic expression (*p*-value = 0.003, period = 12, amplitude = 0.2) [43]. A dramatic reduction of *TZP1/2* after lighting for 10 min, suggested that expression of *TZP1/2* was also regulated by the light signaling pathway (Figure 5A). We continued to monitor the expression profiles for two consecutive days under the constant light conditions. We found that miR530 retained the similar accumulation pattern in Day 1, but the peaked expression shifted about nine hours in Day 2 (Figure 5B,C). Alternatively, the expression of *TZP1/2* remained a single peak expression with slight shifts in Day 1 but displayed disrupted expression patterns in Day 2 (Figure 5B,C). Taken together, these results suggest that the expression of miR530 is potentially under the regulation of circadian rhythm, and both light signaling and circadian clock pathways might be involved in the regulation of *TZP1/2* expression.

## 4. Discussion

Plant miRNAs are critical regulators of gene expression to acclimate to the environmental changes [44]. In this study, we performed a genome-wide identification of drought-responsive miRNAs in the obligate CAM plant *K. marnieriana*. The experimental design focused on the differences between epidermal peels and mesophyll (Figure 1), which revealed potential regulatory miRNAs involved in stomatal regulation and organic acid metabolism. Indeed, the differential expression analysis showed that a small number of miRNAs were commonly regulated in epidermal and mesophyll peels (Figure 3B), suggesting a functional assignment of regulating gene expression by different miRNA families. The genome-wide identification of miRNAs resulted in the discovery of eight potential *Kalanchoë*-specific miRNAs (Figure 2B). Interestingly, we showed that six newly evolved miRNAs were responsive to drought stress (Figure 3C). It will be valuable to investigate the functions and evolution of the *Kalanchoë*-specific miRNAs.

We showed that miR530 was specifically downregulated in epidermal peels but not in mesophyll (Figure 4B) and we also showed that miR530 targeted two TZP genes (*KmTZP1* and *KmTZP2*; Appendix A), whose homologous gene in Arabidopsis controlled the morning-specific growth [41,45]. It has been shown that TZP was an essential regulator of both light and photoperiodic signaling [42,46]. We hypothesize that drought stress might induce the levels of TZP proteins to coordinate the stomatal movement and carboxylation process in CAM species. Moreover, in a recent study of rice, the targeting of TZP homolog by miR530 was revealed to determine the grain yield [47]. Together with our findings in *Kalanchoë*, we postulate that miR530-TZP is an evolutionary conserved regulatory module downstream of the light signaling pathways.

We also found that the expression of miR530 displayed a diurnal rhythm that peaked before dawn (Figure 5). In the obligate CAM plant pineapple (*Ananas comosus*), the diel profiling of miRNA revealed 20 miRNAs belonging to 14 families that were rhythmically expressed in leaves [30]. A member of miR530 clade displayed diel expression, but its target genes were not resolved [30]. The shared diurnal role of miR530 in pineapple and *Kalanchoë* indicates a potential functional convergence of recruiting miRNA-mediated gene regulation in CAM pathway. We showed that the peak expression of miR530 gradually shifted under the constant light conditions (Figure 5), suggesting that the circadian clock was not solely involved in maintaining the diel rhythm. Coincidentally, *KmTZP1/2* were substantially downregulated at the onset of lights (Figure 5). Therefore, we propose that light–dark cycle, as well as the circadian clock, are required to ensure the specific diel expression patterns for miR530 and TZP targets. In recent years, the CRISPR/Cas9 based genome editing tools were shown to be a powerful strategy of generating knockout mutants for studying the functions of the CAM-related genes [22]. Future research can be performed to generate knockout mutants, using CRISPR/Cas9-mediated gene editing, for the miR530 and its target genes to validate their function.

Circadian oscillation of CO_2_ fixation in CAM plants is a distinctive phenomenon that is arguably controlled by a CAM-unique clock [48,49]. Several large-scale transcriptome analyses, using different CAM plants (e.g., agave, ice plant and cacti), have been performed to uncover diel coexpression clusters related to CAM [20,50,51,52]. Our analyses of diurnal expression of miRNAs and their potential targets revealed several critical miRNA-target modules that are associated with CAM-gene expression (Appendix A). Our work provides an additional data source of gene expression underlying CAM clock, and comparative studies of miRNA-targets modules in various CAM plants can be effective methods to uncover the molecular switches underlying convergent C3-to-CAM evolution.

## Figures and Tables

**Figure 1 cells-10-01526-f001:**
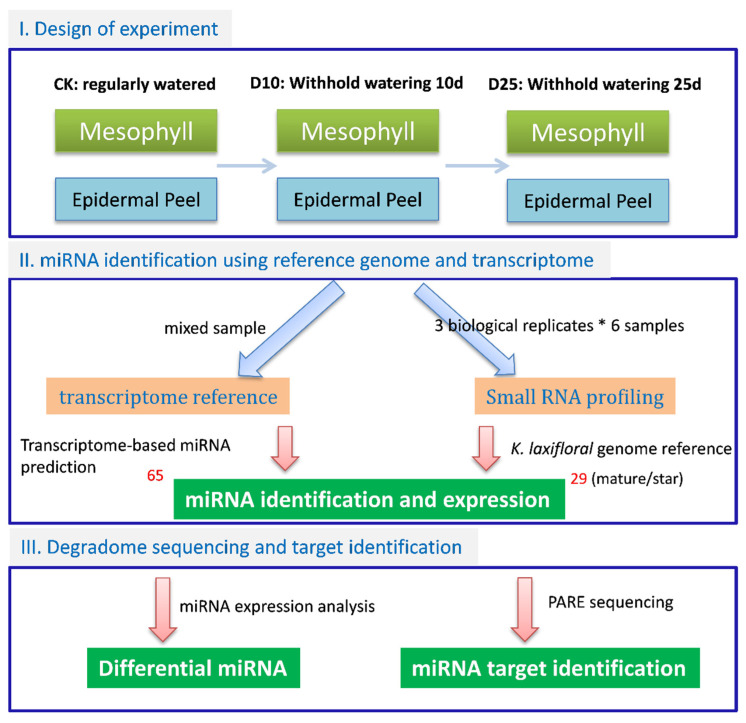
The experimental design for transcriptome sequencing and pipeline of data analysis. Drought treatments (10 and 25 days without watering) were performed using *K. marnieriana* cuttings that were clonal propagated and subsequently sampled for sequencing. The transcriptome and reference genome sequence were used for miRNA identification; the combined miRNA dataset was further analyzed for expression profiling. To reveal the targets of miRNAs, the Parallel Analysis of RNA Ends sequencing was performed using the mixed RNA samples from abovementioned tissues.

**Figure 2 cells-10-01526-f002:**
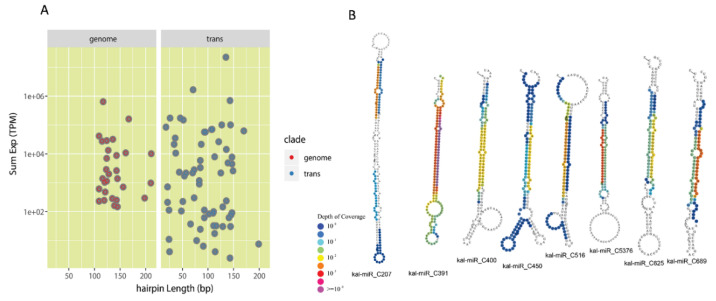
Identification of miRNAs based on small RNA sequencing. (**A**) The distribution of miRNAs by using genome (red dots) and transcriptome (blue dots) as the references, respectively. (**B**) Novel miRNAs identified in *Kalanchoë* that are not homologous in other plant lineages. Different colors of each base indicate the depth of coverage of small RNA sequencing reads that are mapped to the precursor. The plot is made by strucVis version-0.4 (https://github.com/MikeAxtell/strucVis, accessed on 19 June 2018).

**Figure 3 cells-10-01526-f003:**
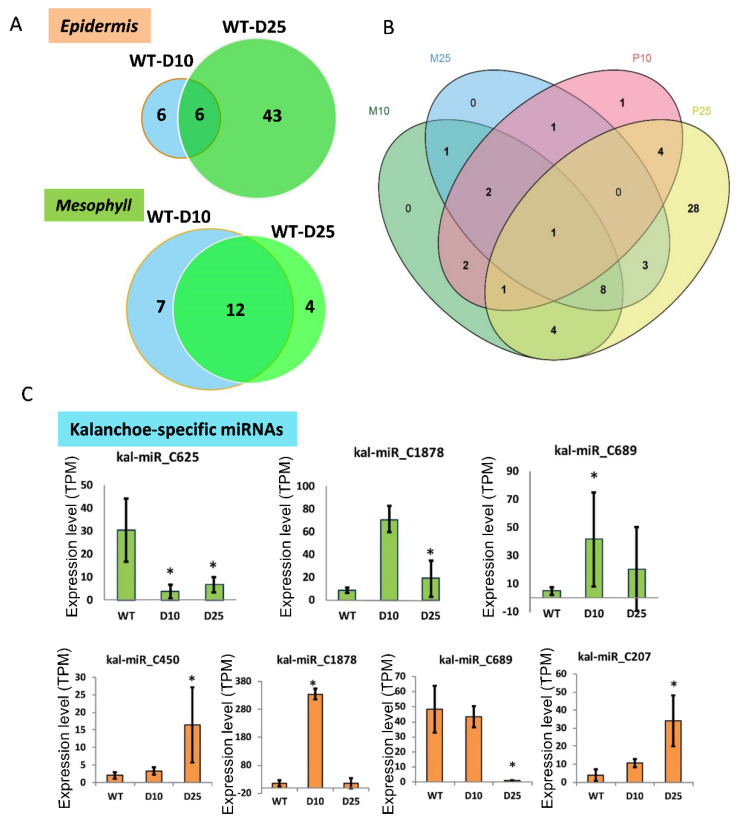
The differentially expressed miRNAs in epidermis and mesophyll in *K. marnieriana*. (**A**) Venn diagram of differentially expressed miRNA under drought conditions in epidermis (orange) and mesophyll (green). The miRNA with total TPM > 10 and 2-fold change (*p*-value < 0.05) were revealed. (**B**) Venn diagram of differentially expressed miRNA in different tissues and drought conditions. (**C**) Differential expression *Kalanchoë*-specific miRNAs. The upper panel indicates the differential expression of miRNAs in mesophyll (green); the lower panel indicates the differential expression of miRNAs in epidermal peels (orange). * indicates significant difference (*p*-value < 0.05) based on the Student’s *t*-test results.

**Figure 4 cells-10-01526-f004:**
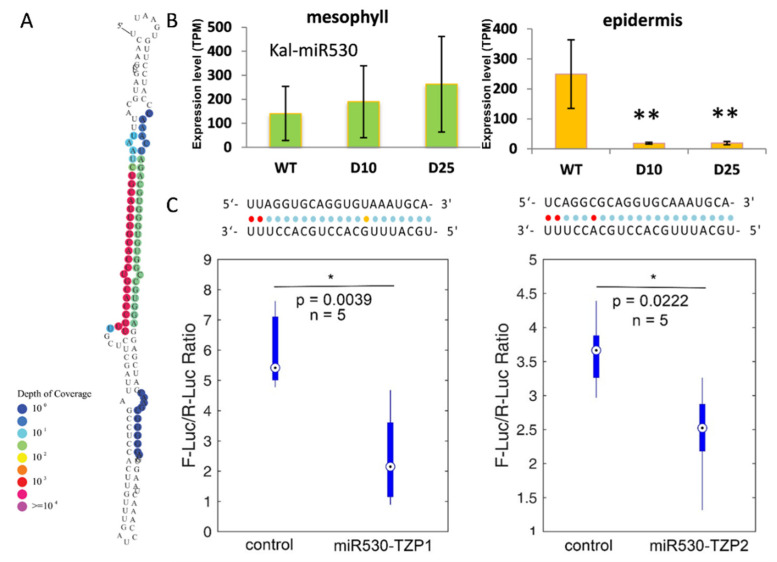
miR530 targets Km*TZP1* and Km*TZP2* in *K. marnieriana*. (**A**) Secondary structure of miR530 precursor in *K. marnieriana*. Different colors indicate the depth of coverage of small RNA sequencing reads that are mapped to the precursor as described before. (**B**) Expression of miR530 mature sequences in epidermis (orange) and mesophyll (green) under the drought treatments. Three biological replicates are used for each quantification and statistical analysis. The Student’s test is performed. Stars indicate significant differences (** indicates *p*-value < 0.01). TPM, Transcripts Per Kilobase of exon model per Million mapped reads. (**C**) Verification of target sites of miR530-*TZP1* (left) and miR530-*TZP2* (right) by the dual-luciferase assay. For each combination of verification, five replicates are used for inoculation as described [31]. The Student’s test is performed. A star indicates significant differences (* indicates *p*-value < 0.05; ** indicates *p*-value < 0.01).

**Figure 5 cells-10-01526-f005:**
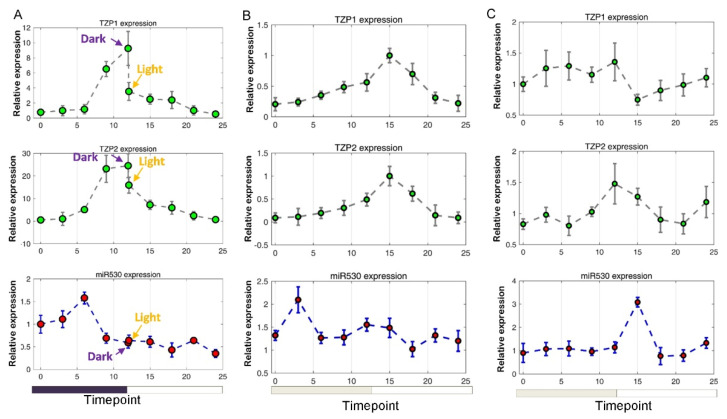
The expression patterns of miR530, *KmTZP1* and *KmTZP2* determined by the circadian clock. (**A**) The diel expression of miR530 and its targets (*KmTZP1* and *KmTZP2*) under the normal light-dark cycle. The interval of time-points is 3 h, and three biological replicates are used for gene expression analysis. The 5 min before and after the light-on are sampled for gene expression analysis (indicated by arrows). (**B**) The diel expression of miR530 and its targets (*KmTZP1* and *KmTZP2*) under constant light conditions, which is performed immediately after the normal light–dark cycle as indicated in (**A**). (**C**) The diel expression of miR530 and its targets (*KmTZP1* and *KmTZP2*) in the second day of constant light conditions. Each panel indicates a sampling cycle of 24 h.

## Data Availability

All data supporting reported results can be found from the links related to the manuscript.

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
