# Peer review of "Transcriptome and Degradome Profiling Reveals a Role of miR530 in the Circadian Regulation of Gene Expression in Kalanchoë marnieriana"

_cells, 2021, doi:10.3390/cells10061526_

Round 1

Reviewer 1 Report

The manuscript by Hu and collaborators reports the identification of miRNAs that are differentially expressed in response to drought in Kalanchoe marianeria, some of which appear to be Kalanchoe-specific. The authors then predict the targets of the miRNAs and more deeply characterize one miRNA-target pair (namely, miR530-TZP1/2). Specifically, they validate the ability of miR530 to target TZP1 and TZP2 in transient assays and they analyze the diel and circadian pattern of expression of miR530 and its targets TZP1 and TZP2.

The findings are overall interesting, but I find that some of the conclusions that can be drawn from the present study are overstated in some parts of the text and I encourage the authors to tone them down. For example, in the abstract, it is stated that “Integrative analysis of differential and diel expression of miRNAs and their targets uncovered novel miRNA-mediated circadian regulation of gene expression. (…) Our work suggests that the miR530-TZPs module plays critical roles of regulating CAM-related gene expression.” This is overstated, given the evidence presented. The study suggests that the circadian clock may impinge on TZP1/2 regulation through miR530 (because of its circadian pattern of expression) but it has not been proven. Also, it is not known what role the miR530-TZPs module may play in CAM-related gene expression (or if it even plays a role at all). Also in the introduction: “We uncovered a novel miR530-TZP module that is involved in the regulation of circadian gene expression.” This is rather inaccurate; the study does not present evidence that the miR530-TZP module is involved in the regulation of circadian gene expression. Additionally, I also find that some important experimental details are missing and should be provided. The manuscript would also undoubtedly benefit from English proofreading.

Below are some additional comments that I encourage the authors to consider:

- Line 156, 162: The term “CAM expression” is confusing and I would recommend rewording. For example, in line 162, instead of “CAM expression is enhanced” I would rather recommend to state that “the content in leaf titratable acids is increased in response to drought” or that “CAM function is enhanced” in response to this treatment.

- Figure 3C, how was this determined? From the sequencing data or confirmed by qPCR? What are the units of the y-axis? Expression level are relative to what?

- Line 2501-251: “This result suggested that the miR530-TZP regulatory module might play crucial roles of directing the diurnal gene expression patterns in Kalanchoë.” In line with my comments above, I think this is overstated.

- Lines 277-291:

  • The experiment does not reveal the role of miR530 in the regulation of circadian gene expression, it just reveals the pattern of expression of miR530, TZP1 and TZP2.
  • A, B and C labeling is missing in figure 5.
  • What are the units of x-axis? How many hours after transfer to constant light should be indicated. How was the experiment performed? Why is it shown in three different graphs?
  • “subjective light conditions” should be reworded to “constant light conditions” or “free running conditions”.
  • By inspecting the data, I am not sure whether miR530 expression can be considered circadian, given the 9 h shift in peak expression in only 1 day. To objectively determine if there are oscillations, an appropriate algorithm such as JTK_cycle (https://openwetware.org/wiki/HughesLab:JTK_Cycle) should be used.
  • “Taken together, these results demonstrate that the expression of miR530 is under the regulation of circadian rhythm, and both light signaling and circadian clock pathways are involved in the regulation of TZP1/2 expression. Again, I would recommend the authors to tone this statement down, because the data presented “suggest” but is insufficient to “demonstrate”.

Author Response

The manuscript by Hu and collaborators reports the identification of miRNAs that are differentially expressed in response to drought in Kalanchoe marianeria, some of which appear to be Kalanchoe-specific. The authors then predict the targets of the miRNAs and more deeply characterize one miRNA-target pair (namely, miR530-TZP1/2). Specifically, they validate the ability of miR530 to target TZP1 and TZP2 in transient assays and they analyze the diel and circadian pattern of expression of miR530 and its targets TZP1 and TZP2.

The findings are overall interesting, but I find that some of the conclusions that can be drawn from the present study are overstated in some parts of the text and I encourage the authors to tone them down. For example, in the abstract, it is stated that “Integrative analysis of differential and diel expression of miRNAs and their targets uncovered novel miRNA-mediated circadian regulation of gene expression. (…) Our work suggests that the miR530-TZPs module plays critical roles of regulating CAM-related gene expression.” This is overstated, given the evidence presented. The study suggests that the circadian clock may impinge on TZP1/2 regulation through miR530 (because of its circadian pattern of expression) but it has not been proven. Also, it is not known what role the miR530-TZPs module may play in CAM-related gene expression (or if it even plays a role at all). 

Also in the introduction: “We uncovered a novel miR530-TZP module that is involved in the regulation of circadian gene expression.” This is rather inaccurate; the study does not present evidence that the miR530-TZP module is involved in the regulation of circadian gene expression. 

Additionally, I also find that some important experimental details are missing and should be provided. The manuscript would also undoubtedly benefit from English proofreading.

#We agree on these comments. In this revision, we have revised the abovementioned statements according to the suggestions. Also, we have provided detailed information on the experiments in the materials and methods. Please find the revised contents in the newly submitted version of manuscript. Thank you.

(1) Specifically, in this revised abstract, we have deleted the sentence "Integrative analysis of differential and diel...". 

(2) The last sentence in abstract has been modified. "Our work suggests that the miR530-TZPs module might play a role of regulating CAM-related gene expression in Kalanchoë."

(3) In introduction, (Line 87-88) the abovementioned sentence is revised.

 "We uncovered that miR530 displays a circadian pattern of expression and targets two tandem zinc knuckle/PLU3 domain encoding genes (TZPs), suggesting a potential role of regulating CAM-related gene expression."

Below are some additional comments that I encourage the authors to consider:

- Line 156, 162: The term “CAM expression” is confusing and I would recommend rewording. For example, in line 162, instead of “CAM expression is enhanced” I would rather recommend to state that “the content in leaf titratable acids is increased in response to drought” or that “CAM function is enhanced” in response to this treatment.

#The comment is taken. We have revised the abovementioned sentence accordingly. 

Line 156, "To assess the drought responses in K. marianeria, we performed the acidity titration measurements between dawn and dusk under different watering conditions."

Line 162, "We found that, under the regularly-watering condition, the difference value (ΔH+) of titratable acid in mature leaves of K. marianeria averaged 72 nmol H+ per kg, and the acidity levels were significantly increased under the 10 days (D10) and 25 days (D25) of drought treatments (Supple. Figure 1C), indicating that the drought treatments enhanced the production of leaf titratable acids."

- Figure 3C, how was this determined? From the sequencing data or confirmed by qPCR? What are the units of the y-axis? Expression level are relative to what?

#Sorry for the overlook. The y-axis is the TPM value of the deep sequencing results of mature miRNA; the "Relative expression level" used in previous version is confusing, and we have revised to "Expression level (TPM)". 

- Line 2501-251: “This result suggested that the miR530-TZP regulatory module might play crucial roles of directing the diurnal gene expression patterns in Kalanchoë.” In line with my comments above, I think this is overstated.

#We agree. We have revised this sentence. "We hypothesized that the miR530-TZP regulatory module might be involved in directing the diurnal gene expression patterns in Kalanchoë."

- Lines 277-291:

#Thank you for the comments. To address the comments and clear up potential confusions, we have revised the paragraph carefully according to the suggestions. Please also find the point-to-point responses below. 

  • The experiment does not reveal the role of miR530 in the regulation of circadian gene expression, it just reveals the pattern of expression of miR530, TZP1 and TZP2.

#The comment is taken. We have rephrased the results to avoid the overstated conclusions. 

  • A, B and C labeling is missing in figure 5.

#Sorry for the overlook. We have added the labels in the revised manuscript. 

  • What are the units of x-axis? How many hours after transfer to constant light should be indicated. How was the experiment performed? Why is it shown in three different graphs?

#Sorry for the confusions. The x-axis indicates the time points with 3-hours interval. The constant light condition is immediately after a normal light-dark cycle in the left panels. 

Each panel from left to right indicates a sampling cycle of 24 hours. To avoid potential misleading, we have revised the figure legends to include the information. 

  • subjective light conditions” should be reworded to “constant light conditions” or “free running conditions”.

#Corrected accordingly. 

  • By inspecting the data, I am not sure whether miR530 expression can be considered circadian, given the 9 h shift in peak expression in only 1 day. To objectively determine if there are oscillations, an appropriate algorithm such as JTK_cycle (https://openwetware.org/wiki/HughesLab:JTK_Cycle) should be used.

#Thank you for the suggestion. The comment is sensible. The JTK-Cycle, based on our understanding, is to measure the period, phase and amplitude of transcript expression under a 24-hour cycle (Hughes et al. 2011); to determine the "rhythmic" expression of transcript, multiple cycles of expression data are appropriate for the analysis. 

We have tested the expression of miR530 using the expression data of three biological replicates; the diel pattern of miR530 expression is consistent under the normal light-dark condition (P-value = 0.0003, PERIOD = 12; LAG = 1.5; AMP = 0.2). But, indeed, the expression peak shifts starting at day 2 of constant light condition. We postulate that the expression of miR530 is regulated by both circadian clock and light signaling pathways. 

Line281-284

" The expression pattern of miR530 under normal light-dark cycle was tested by JTK_cycle, which displayed a rhythmic expression (p-value = 0.003, period = 12, amplitude = 0.2) [43]."

  • “Taken together, these results demonstrate that the expression of miR530 is under the regulation of circadian rhythm, and both light signaling and circadian clock pathways are involved in the regulation of TZP1/2 expression. Again, I would recommend the authors to tone this statement down, because the data presented “suggest” but is insufficient to “demonstrate”.

#The comments are taken. 

Line 289-291 "Taken together, these results suggest that the expression of miR530 is potentially under the regulation of circadian rhythm, and both light signaling and circadian clock pathways might be involved in the regulation of TZP1/2 expression."

Reviewer 2 Report

Overall, the study is well prepared and presented; the experiments seem to be logically planned and visualised and the sequencing data obtained within this study is already available in the database, which may be of use in the future studies of Kalanchoe or more general - CAM plants. There is a point, however, that should be addressed by the authors. I found it peculiar that the authors use the name Kalanchoe marianeria for their plant model throughout the entire manuscript, title included, while in the results (sequencing data deposition, plant image description in Figure F1), they use name Kalanchoe marnieriana. Taking into account that this name (K. marnieriana) is used in the NCBI taxonomy database, sequence database and the literature, including Reference 40 with the Kalanchode phylogenetic analysis, I assume this name is correct. At the same time Kalanchoe marianeria name has 0 (zero) hits even in the general google search. Why the authors use the name of a plant that is not in use anywhere else? This must be explained, justified or corrected, Also, the statements of the authors about the important role of the identified miRNAs in CAM metabolism evolution seem exaggerated e.g.lines 199-202. There is not enough evidence to make such claims based on presented data, since the identified targets are not characterized and there are numerous processes in plant that may be correlated with the diurnal fluctuations of expression. Moreover, miR530 is not Kalanchoe-specific while the current text layout sometimes suggests otherwise.See for example lines 309-315.

Minor corrections:

Figure S1 - reference is missing in the Figure description (B) Table S3- there is a typo in the Arabidopsis word; also - what do the Category numbers mean in this table?

Author Response

Overall, the study is well prepared and presented; the experiments seem to be logically planned and visualised and the sequencing data obtained within this study is already available in the database, which may be of use in the future studies of Kalanchoe or more general - CAM plants. There is a point, however, that should be addressed by the authors. I found it peculiar that the authors use the name Kalanchoe marianeria for their plant model throughout the entire manuscript, title included, while in the results (sequencing data deposition, plant image description in Figure F1), they use name Kalanchoe marnieriana. Taking into account that this name (K. marnieriana) is used in the NCBI taxonomy database, sequence database and the literature, including Reference 40 with the Kalanchode phylogenetic analysis, I assume this name is correct. At the same time Kalanchoe marianeria name has 0 (zero) hits even in the general google search. Why the authors use the name of a plant that is not in use anywhere else? This must be explained, justified or corrected, 

#I am very sorry for the confusion. The comment is sensible. The use of Kalanchoe marianeria is a misfit. In this revision, we have modified it to Kalanchoe marnieriana throughout the manuscript. Thank you

Also, the statements of the authors about the important role of the identified miRNAs in CAM metabolism evolution seem exaggerated e.g.lines 199-202. There is not enough evidence to make such claims based on presented data, since the identified targets are not characterized and there are numerous processes in plant that may be correlated with the diurnal fluctuations of expression. 

#Thank you for the comments. The similar issue is also raised by the reviewer 1. We agree that some statements are overstated. In this revision, we have removed the abovementioned sentence. 

Moreover, miR530 is not Kalanchoe-specific while the current text layout sometimes suggests otherwise. See for example lines 309-315.

#We agree. The abovementioned paragraph is revised. 

Line 308-315 “The genome-wide identification of miRNAs resulted in the discovery of eight potential Kalanchoë-specific miRNAs (Figure 2B). Interestingly, we showed that six newly evolved miRNAs were responsive to drought stress (Figure 3C). It will be valuable to investigate the functions and evolution of the Kalanchoë-specific miRNAs.

We showed that miR530 was specifically down-regulated in epidermal peels but not in mesophyll (Figure 4B) and we also showed that miR530 targeted two TZP genes (KmTZP1 and KmTZP2; Supple. Figure 2B), whose homologous gene in Arabidopsis con-trolled the morning-specific growth [41, 45].”

Minor corrections:

Figure S1 - reference is missing in the Figure description (B) Table S3- there is a typo in the Arabidopsis word; also - what do the Category numbers mean in this table?

#Sorry for the overlook. We have provided the reference in the revised manuscript. In Table S3, the typo is corrected. The category is defined by the number of aligned reads and position of the target sites, which is computed by the CleaveLand 4.0 pipeline. We have added the information in this revised manuscript.
